# Chronic Binge Alcohol and Ovarian Hormone Loss Dysregulate Circulating Immune Cell SIV Co-Receptor Expression and Mitochondrial Homeostasis in SIV-Infected Rhesus Macaques

**DOI:** 10.3390/biom12070946

**Published:** 2022-07-05

**Authors:** Patrick M. McTernan, Robert W. Siggins, Anna Catinis, Angela M. Amedee, Liz Simon, Patricia E. Molina

**Affiliations:** 1Comprehensive Alcohol Research Center, New Orleans, LA 70112, USA; pmcter@lsuhsc.edu (P.M.M.); rsiggi@lsuhsc.edu (R.W.S.); aamede@lsuhsc.edu (A.M.A.); lsimo2@lsuhsc.edu (L.S.); 2Department of Physiology, Louisiana State University Health Sciences Center, New Orleans, LA 70112, USA; acatin@lsuhsc.edu; 3Department of Microbiology, Immunology, and Parasitology, Louisiana State University Health Sciences Center, New Orleans, LA 70112, USA

**Keywords:** SIV, HIV, alcohol, ovariectomy, mitochondria, CD4^+^ T cell, CCR5, CXCR4

## Abstract

Effective antiretroviral therapy (ART) has transitioned HIV to a chronic disease, with more than 50% of people living with HIV (PLWH) being over the age of 50. HIV targets activated CD4^+^ T cells expressing HIV-specific co-receptors (CCR5 and CXCR4). Previously, we reported that chronic binge alcohol (CBA)-administered male rhesus macaques had a higher percentage of gut CD4^+^ T cells expressing simian immunodeficiency virus (SIV) co-receptor CXCR4. Evidence also suggests that gonadal hormone loss increased activated peripheral T cells. Further, mitochondrial function is critical for HIV replication and alcohol dysregulates mitochondrial homeostasis. Hence, we tested the hypothesis that CBA and ovariectomy (OVX) increase circulating activated CD4^+^ T cells expressing SIV co-receptors and dysregulate mitochondrial homeostasis in SIV-infected female rhesus macaques. Results showed that at the study end-point, CBA/SHAM animals had increased peripheral CD4^+^ T cell SIV co-receptor expression, and a lower CD4^+^ T cell count compared to CBA/OVX animals. CBA and OVX animals had altered peripheral immune cell gene expression important for maintaining mitochondrial homeostasis. These results provide insights into how at-risk alcohol use could potentially impact viral expression in cellular reservoirs, particularly in SIV-infected ovariectomized rhesus macaques.

## 1. Introduction

There are an estimated 1.2 million people living with human immunodeficiency virus (PLWH) in the US, with approximately 38,000 new HIV infections occurring yearly [1,2]. PLWH have a 2–3-fold higher prevalence of alcohol use disorder (AUD) than individuals in the general population, and ~17% of PLWH are classified as high-risk drinkers based on Alcohol Use Disorders Identification Test (AUDIT) scores [3,4,5,6,7]. AUD increases the risk of comorbidities, such as neurocognitive impairment, metabolic dysregulation, and cellular and immune aging in PLWH [6]. With the advent of antiretroviral therapy (ART), HIV has become a chronic disease, with ~50% of PLWH being over the age of 50, which includes post-menopausal women, who could be at risk for alcohol-related comorbidities [8].

In 2018, women accounted for 18% of new HIV infections, and HIV-infected women have an earlier onset of menopause, with a mean age range of 46–48 years [9,10,11]. The hormonal changes that occur with menopause alter multiple physiological processes [12]. However, the effect of ovarian hormone loss on HIV pathogenesis and AUD remains to be fully understood. Previous clinical studies have demonstrated that women who underwent clinically indicated ovariectomies (OVX) had a higher percentage of activated peripheral T cells 3 months post-OVX [13]. HIV targets activated CD4^+^ T cells by binding the CD4 receptor and a co-receptor (CCR5 or CXCR4) on the cell surface, leading to depletion of CD4^+^ T cells and an imbalance in the CD4^+^/CD8^+^ ratio [14,15]. Our published data in chronic binge alcohol (CBA)-administered, simian immunodeficiency virus (SIV)-infected male macaques demonstrate that alcohol significantly increases set point plasma viral loads and accelerates progression towards SIV end-stage disease in ART-naïve animals [16], and this was associated with an increased percentage of activated and senescent T cells [17], and an increase in CXCR4-expressing CD4^+^ T cells in the gut [18].

Together with the predominance of target cells, mitochondrial function is also critical for HIV replication [19,20]. Previous studies have demonstrated that inhibitors of oxidative phosphorylation, such as metformin, rotenone, or antimycin A, can decrease HIV replication, preventing T cell depletion [20]. Both preclinical and clinical studies have demonstrated that alcohol alters mitochondrial function by dysregulating mitochondrial biogenesis and increasing oxidative stress [21,22,23]. For example, alcohol decreases PPAR-γ co-activator 1 beta (*PPARGC1B)* expression in skeletal muscle from SIV-infected rhesus macaques, and directly damages mtDNA through the generation of reactive oxygen species (ROS) [24,25,26]. Further, myoblasts isolated from PLWH with high AUDIT scores had an increased mitochondrial content with a decreased oxygen consumption rate (OCR), which is indicative of impaired mitochondrial function, compared to PLWH with low AUDIT scores [27]. These alcohol-mediated changes to mitochondria could result in dysregulated mitochondrial homeostasis, potentially impacting HIV replication. We hypothesized that CBA administration increases the percentage of CD4^+^ T cells expressing SIV co-receptors, and dysregulates mitochondrial homeostasis in ovariectomized, SIV-infected female rhesus macaques. No study to date has investigated the impact of at-risk alcohol use and ovariectomy (to mimic surgical menopause) on HIV levels in viral reservoirs.

## 2. Material and Methods

### 2.1. Animals

All animal experiments were approved by the Institutional Animal Care and Use Committee at the Louisiana State University Health Sciences Center in New Orleans (LSUHSC-NO), LA, which is accredited by Association for Assessment and Accreditation of Laboratory Animal Care International; were performed in accordance with the Guide for the Care and Use of Laboratory Animals; and adhered to National Institutes of Health guidelines for the care and use of animals. The animal housing rooms were maintained on a 12:12-h light:dark cycle, with a relative humidity of 30–70% and a temperature of 64–84 °F (18–29 °C). All animals were fed a standard, commercially formulated nonhuman primate (NHP) diet.

### 2.2. Animal Characteristics and Study Design

Adult (6–9 years old) female Indian rhesus macaques (*Macaca mulatta*) were randomized based on in vitro peripheral blood mononuclear cell (PBMC) SIV infectivity assays as previously described [12,28] (12 & 28 also reported using this cohort of animals). Macaques were administered alcohol intragastrically at a concentration of 30% (*wt*/*vol*) in water (30 min infusion; 5 days/wk; 12–15 g/kg/wk). Peak plasma alcohol concentrations averaged 50–60 mM (~230–280 mg%) 2 h after alcohol initiation. After 3 months of CBA/vehicle (VEH) administration (Pre-SIV), animals were infected intravaginally with SIV_mac251_. CBA or VEH administration continued throughout the study. At viral set-point (2.5 months post-SIV infection), all macaques were initiated on daily subcutaneous injections of 20 mg/kg of Tenofovir (9-(2-phosphonomethoxypropyl) adenine, PMPA) and 30 mg/kg of emtricitabine (FTC), provided by Gilead Sciences Inc. (Foster City, CA, USA). This drug combination and dose have previously been reported to be effective in suppressing viral load and does not result in liver or renal toxicity in SIV-infected macaques [29]. One-month post-initiation of ART (3.5 months post-SIV infection), animals were randomized to either OVX or SHAM surgeries for a total of four treatment groups: VEH/SHAM (*n* = 8), VEH/OVX (*n* = 7), CBA/SHAM (*n* = 7), and CBA/OVX (*n* = 7). Blood was collected in EDTA anti-coagulant tubes and PBMCs were isolated using a Ficoll-paque gradient (Cytiva, Marlborough, MA, USA), counted and cryopreserved at −80 °C. Food consumption was monitored, and animals were provided with Boost/Ensure supplement or PRIMA-Burger (Labserv) when a 10% reduction in daily food intake was observed. Weekly physical examinations included monitoring body weight and measures of general health. Eight months after OVX or SHAM (study endpoint), after an overnight fast, all macaques were euthanized according to the American Veterinary Medical Association’s guidelines. The total time for CBA or VEH administration was 14.5 months, SIV infection was 11.5 months, and all animals were on ART for 9 months (Figure 1).

### 2.3. Plasma Viral Load and PBMC Viral DNA/RNA Measurements

Plasma viral loads and PBMC viral DNA (cell-associated SIV DNA) and RNA (cell-associated SIV RNA) were measured by qPCR and RT-qPCR, respectively, as previously described [30]. Briefly, virions recovered from high-speed centrifugation of 1 mL of plasma were dissolved in 1 mL of Trizol reagent (Life Technologies, Grand Island, NY, USA) and viral RNA purified according to the manufacturer’s protocols. DNA and RNA from PBMCs was extracted using the Qiagen RNeasy Plus Mini Kit according to the manufacturer’s protocol. For analysis of RNA, reverse transcription was performed using the TaqMan Reverse Transcriptase Reagent Kit (Applied Biosystems, Grand Island, NY, USA) for first-strand synthesis. PCR was performed using the, IQ Supermix (BioRad, Hercules, CA, USA) with Primers and probe to SIVgag (forward GCG TCA TTT GGT GCA TTC AC, reverse TCC ACC ACT AGA TGT CTC TGC ACT AT, and FAM-labeled probe with Tamra quencher TGT TTT GCT TCC TCA GTA TGT TTC ACT TTC TCT TCT G). Quantification was determined by comparing the sample Cq value to a DNA or RNA standard curve of 10^1^ to 10^6^ SIV gag copies. The SIV RNA levels in plasma were normalized to copies/mL of plasma. The SIV DNA levels in PBMC were normalized to the cell number, which was determined by qPCR using absolute standards for the single copy gene, RNaseP (RNase P TaqMan #4316844, Applied Biosystems, Waltham, MA, USA), and SIV RNA levels were normalized using absolute standards for ribosomal protein S13 (RPS13) copies, with 3.1 × 10^7^ RPS13 copies/1 × 10^6^ cells used for normalization: forward ATC GCC GCC ATC ATG GGT CG and reverse ATG CGA CCA GGG CAG AGG C, and FAM-labeled probe with Tamra quencher CAA CTT CAA CCA AGT GGG GAC GCT [30]. The limit of detection in all viral quantification assays was 25 copies/10^6^ cells or mL of plasma. Samples with undetectable levels of SIV (<25 copies) were set at 12.5 (halfway between zero and the level of detection). Viral levels were log-transformed for comparisons.

### 2.4. RNA Isolation and Reverse Transcription Quantitative Polymerase Chain Reaction for Mitochondrial-Related Genes

Total RNA was extracted from PBMCs using the miRNeasy Mini Kit (Qiagen, Valencia, CA, USA) and cDNA was synthesized from 1 µg of RNA using the Quantitect reverse transcription kit (Qiagen) according to the manufacturer’s instructions. Custom primers were purchased from Integrated DNA Technologies (Coralville, IA, USA). Final reactions contained cDNA (50 ng), primers (500 nM), and SyBr green (1:2, Quantitect SyBr Green PCR kit, Qiagen) in 20 µL. qPCRs were carried out in duplicate using a CFX96 thermal cycler (Bio-Rad, Hercules, CA, USA), with RPS13 as the housekeeping gene. Results are presented as the fold change expression of target genes versus the control group (VEH/SHAM). The primer sets of mitochondrial-related genes analyzed by RT-qPCR are summarized in Table 1.

### 2.5. Flow Cytometry Analysis

Macaque PBMCs (2 × 10^6^) isolated from peripheral blood were stained to determine T cell, B cell, and monocyte populations. PBMCs were washed with Roswell Park Memorial Institute Media (RPMI)-1640 + 10% FBS and resuspended in 100 μL of PBS. The cells were stained with antibodies (1 μg) for 30 min at room temperature with the following antibody panel (panel 1; Appendix A): AmCyan-Live/Dead (Invitrogen, Grand Island, NY, USA); Qdot 655-CD8; PerCP-CD3 (BD Biosciences, San Jose, CA, USA); PE-CD45; Pacific Blue-CD14; APC-Cy7-CD4; PE-Cy7-CD16 (Biolegend, San Diego, CA, USA); ECD-CD20 (Beckman-Coulter, Brea, CA, USA) FITC-CD38 (StemCell Technologies, Vancouver, BC, Canada); APC-CD66 (Miltenyi Biotech, Auburn, CA, USA). PBMCs were stained for 30 min at room temperature to assess SIV co-receptor expression with the following panel (panel 2; Appendix A): AmCyan-Live/Dead (Invitrogen, Grand Island, NY, USA); Qdot 655-CD8; PerCP-CD3 (BD Biosciences, San Jose, CA, USA); APC-Cy7-CD4 (Biolegend, San Diego, CA, USA); AF488-CXCR4 (R&D Systems); APC-CCR5 (BD Biosciences). Cells were washed with 2 mL of PBS, re-suspended in 300 μL of PBS + 1% paraformaldehyde (PFA), and acquired on an LSRII (BD Biosciences). Data were analyzed using FACS DIVA and FlowJo (TreeStar, Ashland, OR, USA) software.

### 2.6. Statistical Analysis

Data were checked for assumption of normality using the Shapiro–Wilk test. The alpha level set for all statistical analysis was a *p* value ≤ 0.05. SIV load, immune cell distribution, SIV co-receptor expression, and mitochondrial gene expression were analyzed using a 2 (CBA) × 2 (OVX) analysis of variance (ANOVA). When appropriate, post hoc pairwise comparisons were conducted using Tukey’s HSD, except for the mitochondrial gene expression where uncorrected Fischer’s LSD was used for post hoc analysis. Spearson correlations were used to determine whether there was a significant relationship between SIV plasma viral load, PBMC viral RNA, PBMC viral DNA, and immune cell population cell counts (×10^3^/μL). All analyses were performed using GraphPad prism 9.2.0 (San Diego, CA, USA).

## 3. Results

### 3.1. CBA/SHAM Animals Had Higher Plasma Viral Loads, and PBMC Viral DNA and RNA

At the study end-point, two-way ANOVA indicated a significant interaction between CBA and OVX (*p* = 0.05) on plasma viral loads. Post hoc pairwise comparisons confirmed significantly lower plasma viral loads in CBA/OVX animals compared to CBA/SHAM animals (*p* = 0.04; Figure 2A). However, plasma SIV load differences between CBA/SHAM and CBA/OVX animals were also observed prior to OVX (data not shown). There was a significant interaction between CBA and OVX (*p* = 0.02) on SIV DNA levels in PBMCs. Post hoc pairwise comparisons confirmed significantly lower SIV DNA in CBA/OVX compared to CBA/SHAM animals. (*p* = 0.02; Figure 2B). There was a significant interaction between CBA and OVX (*p* = 0.01) on SIV RNA levels in PBMCs. Post hoc pairwise comparisons confirmed significantly higher SIV RNA in CBA/SHAM compared to VEH/SHAM (*p* = 0.004), VEH/OVX (*p* = 0.02), and CBA/OVX animals (*p* = 0.01; Figure 2C).

### 3.2. Immune Cell Phenotype

At the study endpoint, there was a significant interaction between CBA and OVX (*p* = 0.05) on total CD4^+^ cells (×10^3^/µL). Post hoc pairwise comparisons confirmed significantly higher total CD4^+^ cells in CBA/OVX animals compared to CBA/SHAM animals (*p* = 0.02, Figure 3A). There was a main effect of OVX to increase activated CD4^+^CD38^+^ cells (*p* = 0.02; Figure 3B). There were no significant differences in CD8^+^ T cells (×10^3^/μL) (Figure 3C) or activated CD8^+^CD38^+^ T cells (Figure 3D) between the experimental groups. CD20^+^ B cells (×10^3^/μL) did not show any significant differences between the experimental groups (Figure 3E). There was a main effect of OVX to decrease CD14^+^ monocytes (×10^3^/μL; *p* = 0.01; Figure 3F). There were no significant differences in CD14^+^CD16^+^ macrophages (Figure 3G) or CD66^+^ granulocytes (Figure 3H).

### 3.3. Correlation Analysis of Viral Data and Blood Cell Populations

To understand the differences observed in the immune cell populations, Spearman correlation analysis was performed to determine whether the SIV viral load correlated with changes in the peripheral immune cell distribution. CD8^+^CD38^+^ T cells had a significant negative correlation with SIV plasma viral load when all treatment groups were combined (Figure 4A). Further, CD14^+^ monocytes had a significant positive correlation with SIV plasma viral load (Figure 4B). Taken together, these results suggest that animals with a higher viral load had less cytotoxic CD8^+^ T cells, which are responsible for attacking SIV-infected immune cells [31], and higher CD14^+^ monocytes, which can be infected by SIVmac251 [32,33]. The correlation analysis of the SIV viral load and immune cell populations is shown in Appendix A Appendix A.

### 3.4. SIV Co-Receptor Expression

At the study endpoint, there was a significant interaction of CBA and OVX (*p* = 0.03) on SIV co-receptor and CCR5 expression on CD4^+^ T cells. Post hoc pairwise comparisons confirmed that CCR5 expression on CD4^+^ T cells from CBA/OVX animals was significantly lower compared to VEH/SHAM (*p* = 0.02), VEH/OVX (*p* = 0.05), and CBA/SHAM (*p* = 0.01) (Figure 5A). There was a significant interaction of CBA and OVX on SIV co-receptor and CXCR4 expression on CD4^+^ T cells (*p* = 0.05). Post hoc pairwise comparisons confirmed that C*X*CR4 expression on CD4^+^ T cells from CBA/OVX animals was significantly lower compared to VEH/SHAM (*p* = 0.01), VEH/OVX (*p* = 0.03), and CBA/SHAM (*p* = 0.01) animals (Figure 5B).

### 3.5. PBMC Mitochondrial-Related Gene Expression

At the study endpoint, there was a significant interaction of CBA and OVX on PBMC Mitofusion 2 (*MFN2)* gene expression (*p* = 0.01), important for mitochondrial fusion and mitochondrial repair. Post hoc pairwise comparisons confirmed increased *MFN2* gene expression in PBMCs from CBA/SHAM compared to VEH/SHAM (*p* = 0.01) and CBA/OVX (*p* = 0.01) animals (Figure 6A). There was a main effect of OVX to decrease PBMC superoxide dismutase 2 (*SOD2)* gene expression (*p* = 0.05; Figure 6B), important for decreasing oxidative stress within mitochondria. There was a significant interaction of CBA and OVX on GA-binding protein subunit beta-1 (*GABP1)* gene expression (*p* = 0.01), important in regulating the antioxidant defense. Post hoc pairwise comparisons confirmed significantly increased PBMC *GABP1* gene expression in VEH/OVX (*p* = 0.03) compared to VEH/SHAM (*p* = 0.03) and CBA/OVX (*p* = 0.04) animals (Figure 6C). In the current study, younger animals were used, and this is a limitation, as aging has been shown to impact mitochondrial homeostasis [34]. Therefore, further studies will be performed to functionally validate these changes in mitochondrial gene expression by CBA and OVX in older animals.

## 4. Discussion

We sought to determine whether the combination of CBA and OVX altered immune cell phenotypes and mitochondria-related gene expression in SIV-infected rhesus macaques. We observed that CBA/OVX animals had decreased PBMC viral DNA and RNA levels but higher absolute numbers of CD4^+^ T cells compared to CBA/SHAM animals. Additionally, CD4^+^ T cells of CBA/OVX animals had decreased SIV co-receptor expression, which partially contributed to the reduced SIV DNA and RNA levels in these animals. Further, the combination of CBA and OVX altered the expression of genes related to mitochondrial repair and regulation of oxidative stress in peripheral mononuclear cells, which is important for SIV replication.

HIV targets activated CD4^+^ T cells by binding to the CD4 receptor and a co-receptor (CCR5 or CXCR4) [14,15]. CD4^+^ T cell SIV co-receptor CXCR4 expression had a positive significant correlation with plasma SIV viral load and PBMC viral DNA while SIV co-receptor CCR5 expression had a positive significant correlation with PBMC viral DNA and RNA (Appendix A Appendix A). Though SIV_mac251_ uses CCR5 and does not utilize CXCR4, we were interested to know the impact of CBA and OVX on the expression of both co-receptors as they are used by other strains of SIV and HIV [35]. Activated CD4^+^ T cells that express a co-receptor are more susceptible to viral infection and actively produce more virus than quiescent cells, leading to higher viral loads [36]. Previous clinical studies have demonstrated that women who underwent clinically indicated ovariectomy had increased activated T cells in peripheral blood, and pre-clinical studies showed that CBA-administered macaques had increased proliferating gut CD4^+^ T cells expressing the viral co-receptor CXCR4 [18]. Therefore, we predicted that CBA and OVX would synergistically increase HIV target cells.

At the study end-point, CBA/SHAM animals had a significantly higher plasma viral load and PBMC viral RNA (expressed SIV virus) and DNA (SIV proviral loads) compared to VEH/SHAM animals. However, neither VEH/OVX nor CBA/OVX animals had increased plasma viral loads or PBMC RNA and DNA compared to VEH/SHAM animals. Further, correlation analysis suggested decreased activated cytotoxic CD8^+^ T cells, which are important in eliminating SIV-infected cells [31], and increased CD14^+^ monocytes, which are potential targets of SIVmac251 [32,33], with an increasing SIV virus load independent of the experimental conditions. Despite these favorable conditions for SIV viral replication, we predict that the decreased expression of SIV co-receptors on CD4^+^ T cells in CBA/OVX animals could explain the lower SIV viral loads in these animals.

HIV replication relies on immunometabolism [37,38,39]. Mitochondrial respiration is critical for HIV replication, as administration of metformin decreased viral loads in humanized HIV-infected mice [20]. Alcohol dysregulates gene expression implicated in mitohormesis, a response to stress to restore the health and viability of mitochondria [24,40]. CBA/SHAM animal gene expression data are suggestive of a mitohormetic response for mitochondrial repair and antioxidant defenses, but these responses were not observed in CBA/OVX animals. CBA/SHAM animals had increased *MFN2* expression, important for mitochondrial fusion and mitochondrial repair, while CBA/OVX had similar *MFN2* expression compared to VEH/SHAM animals. Moreover, there was a main effect of ovariectomy to decrease *SOD2* expression, which is important for decreasing oxidative stress within mitochondria. Further, VEH/OVX animals had significantly higher expression of *GABP1*, important in regulating the antioxidant defense, compared to both VEH/SHAM and CBA/OVX, indicating a lack of response to oxidative stress in CBA/OVX animals. Overall, CBA/OVX animals had a similar expression of genes implicated in mitochondrial homeostasis as VEH/SHAM, and we posit that the lower levels of virus integrated and expressed in PBMCs could result in a less proinflammatory milieu, and thus decreased mitochondrial stress [41].

This study has some limitations. The sample size is small; however, the studies are in non-human primates that have biological variability similar to humans, and the consistent and statistically significant changes observed are biologically relevant. We used young animals, and this is a limitation to understanding the impact of CBA and OVX on mitochondrial homeostasis, as aging has been shown to impact mitochondrial homeostasis [34]. Moreover, the studies did not include functional validation of the changes in mitochondrial-related gene expression. Future studies will determine mitochondrial volume, bioenergetic function, and mitochondrial ROS-related protein expression and how the decreased mitohormetic response affects SIV viral dynamics within older animals.

## 5. Conclusions

Our findings advance the understanding of the impact of at-risk alcohol use and OVX on SIV infection and expression. This study is the first to show that CBA-administered/OVX animals had decreased CD4^+^ T cell SIV co-receptor expression (CCR5 and CXCR4), which is important for SIV infection of CD4^+^ T cells. This unexpected decrease in the expression of CCR5 and CXCR4, which was contrary to our original hypothesis, potentially explains the lower SIV viral loads despite the presence of a greater number of activated CD4^+^ T cells. Further, CBA/OVX animals had altered PBMC gene expression associated with mitochondrial homeostasis compared to CBA/SHAM animals, and we speculate that this could be due to decreased viral stress in CBA/OVX animals. Further studies are needed to investigate the mechanisms of CBA- and OVX-mediated changes on CD4^+^ T cell bioenergetics in older animals and whether they contribute to increased HIV/SIV levels. Overall, these studies provide insights into how at-risk alcohol use modulates viral expression in reservoirs, particularly in HIV-infected, post-menopausal women.

## Figures and Tables

**Figure 1 biomolecules-12-00946-f001:**
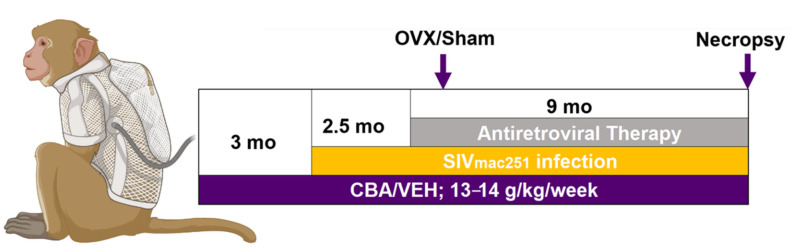
Macaque study design. Adult (6–9 year old) female Indian rhesus macaques (*Macaca mulatta*) were administered chronic binge alcohol (CBA) at a concentration of 30% (wt/vol) in water (30 min infusion; 5 days/wk; 12–15 g/kg/wk) intragastrically or isovolumetric water (vehicle; VEH). After 3 months of CBA/VEH administration (pre-simian immunodeficiency virus (SIV)), all animals were infected intravaginally with SIV_mac251_. CBA/VEH administration continued throughout the study. At viral set-point (2.5 mo post-SIV infection), all macaques were initiated on antiretroviral therapy (ART). One month post-initiation of ART, animals were randomized for either ovariectomy (OVX) or SHAM surgeries and humanely euthanized at 8 months following OVX or SHAM (Necropsy). The total time for CBA/VEH administration was 14.5 months, SIV infection 11.5 months, and ART 9 months. Samples from necropsy were used for this study.

**Figure 2 biomolecules-12-00946-f002:**
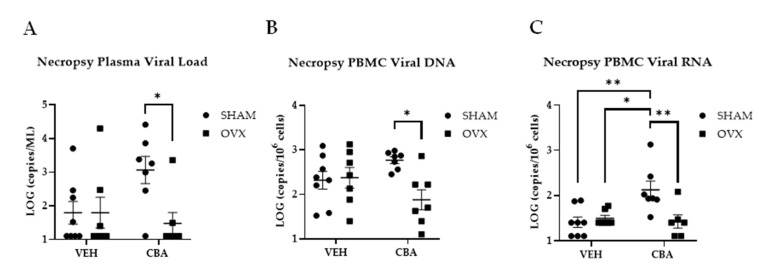
Plasma and PBMC viral loads. (**A**) CBA/SHAM animals had a higher plasma viral load compared to CBA/OVX animals. (**B**) PBMC viral DNA was higher in CBA/SHAM compared to CBA/OVX animals. (**C**) PBMC viral RNA was higher in CBA/SHAM compared to all other groups. A LOG value of 1.1 was used for SIV-infected samples that had undetected virus. Vehicle (VEH)/sham surgery (SHAM) (*n* = 8), VEH/ovariectomy (OVX) (*n* = 7), chronic binge alcohol (CBA)/SHAM (*n* = 7), and CBA/OVX (*n* = 7). Two-way ANOVA and Tukey’s multiple comparison test. * *p* ≤ 0.05, ** *p* ≤ 0.01.

**Figure 3 biomolecules-12-00946-f003:**
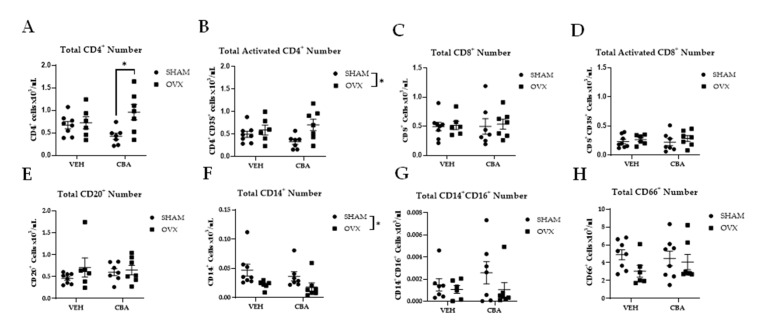
Immune cell distribution. (**A**) CBA/OVX had significantly higher CD4^+^ T cells compared to CBA/SHAM. (**B**) There was a main effect of OVX to increase activated CD4^+^ (CD38^+^) T cell numbers (asterisk represents the main effect of OVX). (**C**) There were no statistically significant differences in CD8^+^ T cell and (**D**) activated CD8^+^ (CD38^+^) T cell numbers between groups. (**E**) There were no statistically significant differences in CD20^+^ cells between groups. (**F**) There was a main effect of ovariectomy to decrease CD14^+^ monocytes (asterisk represents main effect of OVX). (**G**) There were no statistically significant differences in CD14^+^CD16^+^ monocytes or (**H**) CD66^+^ granulocytes. Vehicle (VEH)/sham surgery (SHAM) (*n* = 8), VEH/ovariectomy (OVX) (*n* = 7), chronic binge alcohol (CBA)/SHAM (*n* = 7), and CBA/OVX (*n* = 7). Two-way ANOVA and Tukey’s multiple comparison test. * *p* ≤ 0.05.

**Figure 4 biomolecules-12-00946-f004:**
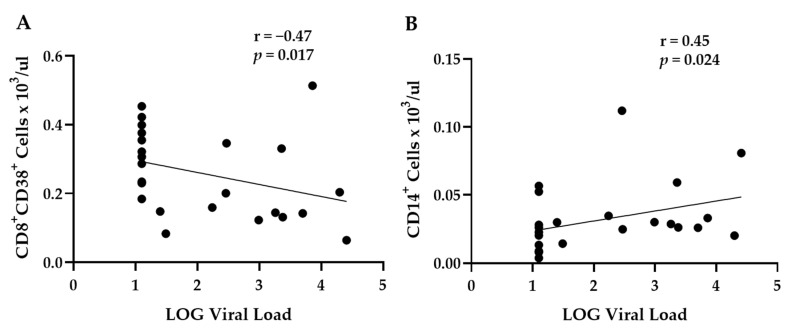
Spearman correlations of viral load with immune cell populations. (**A**) Viral load had a significant negative correlation with activated CD8^+^CD38^+^ T cell counts. (**B**) Viral load had a positive significant correlation with CD14^+^ monocyte counts. A LOG value of 1.1 was used for SIV-infected samples that had undetected virus. Vehicle (VEH)/sham surgery (SHAM) (*n* = 8), VEH/ovariectomy (OVX) (*n* = 7), chronic binge alcohol (CBA)/SHAM (*n* = 7), and CBA/OVX (*n* = 7). Two-way ANOVA and Tukey’s multiple comparison test. *p* ≤ 0.05.

**Figure 5 biomolecules-12-00946-f005:**
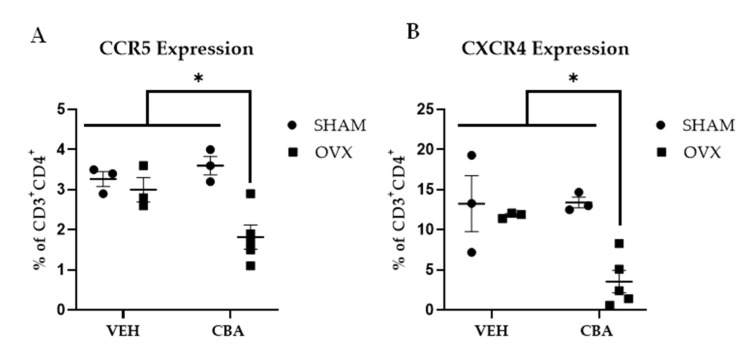
SIV co-receptor expression. CBA/OVX has significantly lower CCR5 and CXCR4 expression compared to all other experimental groups. (**A**). *CCR5* expression is significantly decreased in CBA/OVX compared to all other experimental groups. (**B**). CXCR4 expression is significantly decreased in CBA/OVX compared to all other experimental groups. Vehicle (VEH)/sham surgery (SHAM) (*n* = 3), VEH/ovariectomy (OVX) (*n* = 3), chronic binge alcohol (CBA)/SHAM (*n* = 3), and CBA/OVX (*n* = 5). Two-way ANOVA and Tukey’s multiple comparison test. * *p* ≤ 0.05.

**Figure 6 biomolecules-12-00946-f006:**
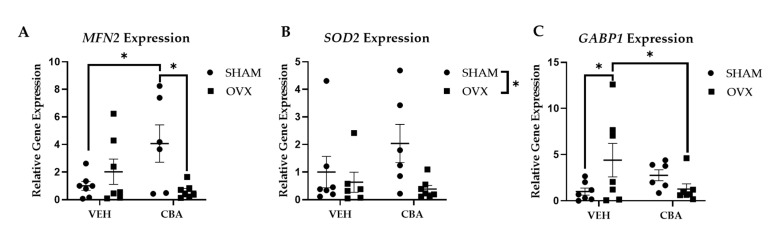
Expression of genes implicated in mitochondrial homeostasis. (**A**). CBA/SHAM had significantly higher Mitofusion 2 (*MFN2*) expression compared to CBA/OVX (**B**). There was a main effect of OVX to decrease superoxide dismutase 2 *(SOD2*) expression (asterisk represents the main effect of OVX). (**C**). VEH/OVX had increased GA-binding protein subunit beta-1 (*GABP1*) expression compared to the VEH/SHAM and CBA/OVX groups. Vehicle (VEH)/sham surgery (SHAM) (*n* = 8), VEH/ovariectomy (OVX) (*n* = 7), chronic binge alcohol (CBA)/SHAM (*n* = 7), and CBA/OVX (*n* = 7). Two-way ANOVA and uncorrected Fischer’s LSD. * *p* ≤ 0.05.

**Table 1 biomolecules-12-00946-t001:** List of gene targets analyzed by RT-qPCR.

Gene Symbol	Gene Targets
**Mitochondrial-Related Genes and Primer Sequences**	
*MFN2—*Mitofusin 2	Forward: CTGTGCTGGTGGATGATTAC Reverse: CCCAGTCCTTCCTCTATGT
*SOD2—*Superoxide Dismutase 2	Forward: GACAAACCTCAGCCCTAATG Reverse: CCGTCAGCTTCTCCTTAAAC
*GABP1—*GA-Binding Protein Subunit Beta-1	Forward: GTGCAATCTGCTACACCTAC Reverse: GATAGAAGCTCACCTGGGA
**House Keeping Gene**	
*RPS13—*Ribosomal Protein S13	Forward: TTGCTCATCTTCCTGA Reverse: CCGGCTCTCTATCAGAATCA

## Data Availability

The data presented in this study are available on request from the corresponding author.

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
