# Peer review of "Chronic Binge Alcohol and Ovarian Hormone Loss Dysregulate Circulating Immune Cell SIV Co-Receptor Expression and Mitochondrial Homeostasis in SIV-Infected Rhesus Macaques"

_biomolecules, 2022, doi:10.3390/biom12070946_

Round 1

Reviewer 1 Report

Dear Authors, 

The manuscript is the continuation of previously published papers on the same field. The paper is well written and structurally organized.

However, I have some questions and suggestions.

1- The Authors correctly mention that antiretroviral therapy is able to decrease alone the viral load. Did the Authors have data concerning the same procedure of ovariectomy without antiretroviral therapy. Indeed it should be interesting if OVX has some effect also without the presence of antiretroviral therapy.

2- line 78, page 2: The references 12 and 28 should be in square brackets.

3- VEH: please use the full-length word (vehicle, I suppose) for the first time.

4- Figure 3, panels B and F: Please, explain why the asterisk (*) is close to the figure legend. The same for panel B, figure 6.

Author Response

We thank the reviewer for their edits and suggestions. Responses to the reviewers edits are shown below:

1- The Authors correctly mention that antiretroviral therapy is able to decrease alone the viral load. Did the Authors have data concerning the same procedure of ovariectomy without antiretroviral therapy. Indeed it should be interesting if OVX has some effect also without the presence of antiretroviral therapy

Response: The reviewer brings up a good point for the use of a non-ART group within the study, but our goal was to model our PLWH cohort (> 94% ART adherence and 75% with viral loads < 50 copies/ml) using a female SIV-infected macaque model. 

2- line 78, page 2: The references 12 and 28 should be in square brackets.

Response: We thank the reviewer for their comment.  We have made this correction in the text on page 2, Lines 78-79.

3- VEH: please use the full-length word (vehicle, I suppose) for the first time.

Response: We have made this correction in the text on page 2. Lines 81 (Shown below).

After 3 months of CBA/Vehicle (VEH) administration (Pre-SIV), animals were infected intravaginally with SIVmac251. CBA or VEH administration continued throughout the study."

4- Figure 3, panels B and F: Please, explain why the asterisk (*) is close to the figure legend. The same for panel B, figure 6.

Response: The asterisks represent main effects of ovariectomy for panels B and F (figure 3) and panel B (figure 6). We have clarified this in the figure legends on page 6, lines 190-191 and 192-193 and page 8, lines 243-244 (shown below).

“There was a main effect of OVX to increase activated CD4+ (CD38+) T cell numbers (asterisk represents main effect of OVX).”

Reviewer 2 Report

Millions of people in the world are infected with Human Immunodeficiency Virus (HIV) and every minute there is a new infection with the virus. The advent of antiretroviral therapy (ARV)has turned HIV-infection into a chronic disease, and many of the people living HIV in developed countries are over 50 years old, while the prevalence of alcohol-related disorders in them is 2-3 times higher than in people in the general population. Therefore, the study which attempts to investigate the influence of two factors: social and physiological, on the development of HIV-infection in women, is undoubtedly relevant.

  The first factor is social, associated with chronic alcohol intake, the second is physiological, associated with age-related changes in women during menopause. The authors presented data on how alcohol consumption and menopause conditions, individually and in combination, it modulates the expression of the virus, affects the immune status and functioning of mitochondria against the background of ARV. Female Indian rhesus monkeys were chosen as the model. Ovariectomy (OVX) was used to create conditions close to menopause. Results showed that at study end-point, that chronic binge alcohol (CBA)-administered animals had increased peripheral CD4+T cell SIV co-receptor expression, and a lower CD4+T cell count compared to CBA/OVX animals. CBA and OVX animals had altered peripheral immune cell gene expression important for maintaining mitochondrial homeostasis.

   Unfortunately, the obtained results are difficult to interpret and draw certain conclusions, since only treated animals were used in the work. Despite the fact that the functioning of the immune system of monkeys is basically similar to that of humans, there are differences in the features of replication of SIV and HIV strains and their effect on organisms. The interaction of the human immune system with HIV has specific features that are determined by the complex structure of many host genes and their metabolism, and the multifaceted ability of the virus genome to change and use the biological mechanisms of the host for its own purposes. It seems important to study the same groups of animals without treatment.

There is strong evidence that alcohol is associated with an increase in the incidence of HIV, tuberculosis and pneumonia and a deterioration in treatment outcomes through both behavioral and biological mechanisms. HIV-infected and treated patients may have impaired mitochondrial and metabolic profiles, but the specific contribution of viral or therapeutic toxicity remains unclear. Menstrual and hormonal disorders in HIV-infected women are multifactorial and lead to the early onset of menopause. The mechanism of early menopause is insufficiently studied, but there is evidence that the immune system affected by menopause leads to a decrease in the number of CD4 cells (<200 cells /mcl) and further progression of the disease. These results highlight the importance of proactive assessment of status and symptoms in this patient population to ensure appropriate support and treatment. This is very important if we want to help women living with HIV optimize their health as they reach middle age and older.

   Based on the importance for healthcare of prognostic data on the peculiarities of the development of HIV infection and the improvement of approaches to its treatment in various groups of patients, the study conducted for the first time can serve as a basis for expanding and comprehensive joint study of the problems raised.

Author Response

We thank the reviewer for their comments. The reviewer brings up a good point on performing this study in the absence of ART. In our study, all female macaques did receive ART after 2.5 months of SIV infection. In the US, ART adherence among PLWH is ~74% and in our clinical cohort >94% are ART adherent. Hence, we wanted to model the preclinical studies to determine the interactions of alcohol and/or ovariectomy on immune cell dynamics in SIV-infected ART-treated macaques. Additionally, our previous publications demonstrate that many of the observed alcohol-mediated effects are irrespective of ART treatment (Ford et al. 2016, Molina et al. 2014).

This manuscript is a resubmission of an earlier submission. The following is a list of the peer review reports and author responses from that submission.